# Non-canonical NOTCH3 signalling limits tumour angiogenesis

Shuheng Lin[1], Ana Negulescu[1], Sirisha Bulusu[1], Benjamin Gibert[1], Jean-Guy Delcros[1], Benjamin Ducarouge[1], Nicolas Rama[1], Nicolas Gadot[2], Isabelle Treilleux[2], Pierre Saintigny[2], Olivier Meurette[1,*] & Patrick Mehlen[1,2,*]

Notch signalling is a causal determinant of cancer and efforts have been made to develop targeted therapies to inhibit the so-called canonical pathway. Here we describe an unexpected pro-apoptotic role of Notch3 in regulating tumour angiogenesis independently of the Notch canonical pathway. The Notch3 ligand Jagged-1 is upregulated in a fraction of human cancer and our data support the view that Jagged-1, produced by cancer cells, is inhibiting the apoptosis induced by the aberrant Notch3 expression in tumour vasculature. We thus present Notch3 as a dependence receptor inducing endothelial cell death while this pro-apoptotic activity is blocked by Jagged-1. Along this line, using Notch3 mutant mice, we demonstrate that tumour growth and angiogenesis are increased when Notch3 is silenced in the stroma. Consequently, we show that the well-documented anti-tumour effect mediated by γ-secretase inhibition is at least in part dependent on the apoptosis triggered by Notch3 in endothelial cells.

---

[1] Apoptosis, Cancer and Development Laboratory—Equipe labellisée 'La Ligue', LabEx DEVweCAN, Centre de Recherche en Cancérologie de Lyon, INSERM U1052-CNRS UMR5286, Université de Lyon, Centre Léon Bérard, 69008 Lyon, France. [2] Department of Translational Research and Innovation, Centre Léon Bérard, 69008 Lyon, France. * These authors contributed equally to this work. Correspondence and requests for materials should be addressed to O.M. (email: olivier.meurette@lyon.unicancer.fr) or to P.M. (email: patrick.mehlen@lyon.unicancer.fr).

Tumour angiogenesis has been considered as an attractive target for cancer therapy for more than forty years. However, clinical results using drugs targeting tumour angiogenesis are inconsistent and often disappointing[1]. Most anti-angiogenic therapies target the vascular endothelial growth factors (VEGFs) signalling pathways, in which VEGFs activate VEGF receptors (VEGFRs) on endothelial cells to regulate vascular growth in both developing tissues and growing tumours. Notch signalling is a major regulator of these processes. Four Notch receptors (Notch1-4) have been described in mammals. Notch receptors are single-pass type I trans-membrane non-covalently linked heterodimer coded by a single precursor, which is cleaved by furins. The Notch pathway activation follows the binding of the transmembrane ligands of the Delta/Serrate/LAG-2 (DSL) family, Delta-like and Jagged to Notch receptors. In mammals, three Delta-like ligands (Dll1, Dll3 and Dll4) and two Jagged ligands (Jag-1 and Jag-2) have been identified. The well-described so-called 'canonical pathway' depends on a strictly controlled proteolytic cascade induced by ligand binding: an S2 cleavage by metalloproteases followed by an S3 cleavage mediated by a presenilin-γ-secretase complex. These proteolytic cleavages release the intracellular domain of the Notch receptor (NICD), which then translocates into the nucleus to mediate target genes activation[2].

Notch signalling has been implicated in cancer, with observed genetic alterations in a large number of hematopoietic and solid tumours[3]. As the presenilin-γ-secretase complex activity is necessary for the activation of the canonical signalling pathway, γ-secretase inhibitors such as DAPT (N-[N-(3,5-difluorophenacetyl)-L-alanyl]-S-phenylglycine t-butyl ester) derivatives have been proposed as targeted therapies for treatment of pathologies such as T-cell acute lymphoblastic leukemia. However, such therapeutic approaches have so far been limited due to intestinal toxicity[4]. Other approaches to inhibit the Notch canonical pathway are thus in development with strategies including antibodies raised specifically against individual Notch receptors[5,6].

Notch signalling is also a major regulator of angiogenesis as Dll4-mediated Notch activation controls the expression of the VEGFRs and therefore limits endothelial cells sprouting and proliferation[7,8]. However, whereas the role of Notch signalling is well described in developmental angiogenesis, its role in tumour angiogenesis is not clearly understood. In vitro, Notch inhibition has been shown to induce endothelial cell death[9] as well as vascular sprouting[10]. In vivo, Notch inhibition using chemical inhibitors or Notch1 ectodomain is generally associated with endothelial cell death and reduced vascularization[11–13]. In contrast, anti-ligand approaches such as anti-Dll4 treatments produces non-productive angiogenesis through increased endothelial cells sprouting[14]. These paradoxical observations could suggest that the role of Notch in tumour angiogenesis cannot be completely explained by canonical Notch signalling. In contrast to other Notch receptors, Notch3 expression is restricted to the vasculature in physiological condition. Notch3 mutations are associated with CADASIL[15] (cerebral autosomal dominant arteriopathy with subcortical infarcts and leukoencephalopathy) and Notch3 knockout mice are more susceptible to ischemic stroke[16] whereas they are less susceptible to pulmonary hypertension[17]. These studies show that even if Notch3 mutant mice have no major phenotype in developmental angiogenesis, Notch3 is involved in pathological angiogenesis. However, its role in tumour angiogenesis has never been studied. In the disorganized tumour vasculature, tumour endothelial cells show a different phenotype than normal endothelial cells[18]. Interestingly, Notch3 has been shown to be upregulated in human lung cancer-associated endothelial cells[19] and this led us to evaluate the role of Notch3 in endothelial cell in cancer development. While analysing the importance of Notch3 in the stroma during tumour progression, we observed an unexpected pro-apoptotic activity of Notch3. We describe Notch3 as a dependence receptor in endothelial cells. Such receptors that include the netrin-1 receptors DCC and UNC5H (ref. 20) or the Hedgehog receptors Ptc and CDON[21,22] share the ability to actively transduce a death signal in settings of ligand limitation, thus creating a state of cellular dependence to the presence of ligand for cell survival. This pro-apoptotic activity has been proposed to act as a negative constrain for tumour progression by controlling cancer cell death[23,24]. We propose here that Notch3 by acting as a dependence receptor in endothelial cells regulate tumour angiogenesis by regulating endothelial cell death.

## Results

**Notch3 is expressed in tumour associated endothelial cells.** We first investigated Notch3 expression in a small panel of human lung cancers by immunohistochemistry. In all the studied samples (11 adenocarcinoma (ADC) and 10 squamous cell carcinoma (SCC)), the expression of Notch3 was very strong in the vasculature (Supplementary Fig. 1a). Conversely, the cancer cell expression of Notch3 was very heterogeneous between patients but also within the same patient (Supplementary Fig. 1a). SCC showed the strongest Notch3 expression in the cancer cells, however, only a small fraction of patients showed nuclear expression (4/10 for SCC and 2/11 for ADC) (Supplementary Fig. 1a,b). The role of Notch signalling and in particular Notch3 in the epithelial compartment of tumours and more specifically of non-small cell lung cancers has been extensively studied[25,26]. However, Notch3 implication in tumour vasculature has not been addressed. We thus focused on the vascular expression of Notch3 in these patients. In the patients for whom we could observe histological normal peritumoral tissue, we noticed that the expression of Notch3 was localized, as described previously[17], in the vascular smooth muscle cells or in the mural cells of smaller vessels (Fig. 1a). However, in the malignant part, we could observe Notch3 expression in the endothelial cells (EC) (Fig. 1a). This prompted us to investigate a possible role of this aberrant expression of Notch3 in tumour endothelial cells. To study this role, we first assessed whether this aberrant expression was also observed in mouse model of lung cancers. We first purified EC from lung adenomas in the $Kras^{+/G12D}$ hit and run mice model characterized previously[27]. Whereas in wild-type mice or in the healthy part of lung from $Kras^{+/G12D}$ mice, no or little expression of Notch3 was detected in EC-enriched fraction, we observed an over-expression of Notch3 in the EC-enriched fraction from the tumour nodules (Fig. 1b). We next used the LacZ reporter to monitor Notch3 expression in the Notch3 LacZ knock-in mice described previously[16]. We confirmed, in this model, that the Notch3/LacZ mRNA fusion was expressed to a similar amount than the wild-type allele (Supplementary Fig. 2a). We also used an anti-β-galactosidase antibody to check the staining of the LacZ enzymatic reaction and confirmed expression in the smooth muscle cells in healthy lungs (Supplementary Fig. 2b). As described by others, we observed that the expression of Notch3 was restricted to mural cells and Notch3 was absent in endothelial cells in normal vasculature. Notch3 was indeed mostly associated to α-smooth-muscle actin (αSMA) expressing cells surrounding big vessels and to a lesser extend to NG2 (neural/glial antigen 2) expressing mural cells in smaller vessels as seen in lungs from $Notch3^{+/LacZ}$ mice. We then looked at Notch3 expression in the adenocarcinoma from $Kras^{+/G12D}$-$Notch3^{LacZ/+}$ mice. We confirmed the data obtained by purifying tumour-associated endothelial cells. Indeed, the LacZ staining is detected in the tumour and in healthy lung, the LacZ staining is not associated with ERG staining—that is, ERG

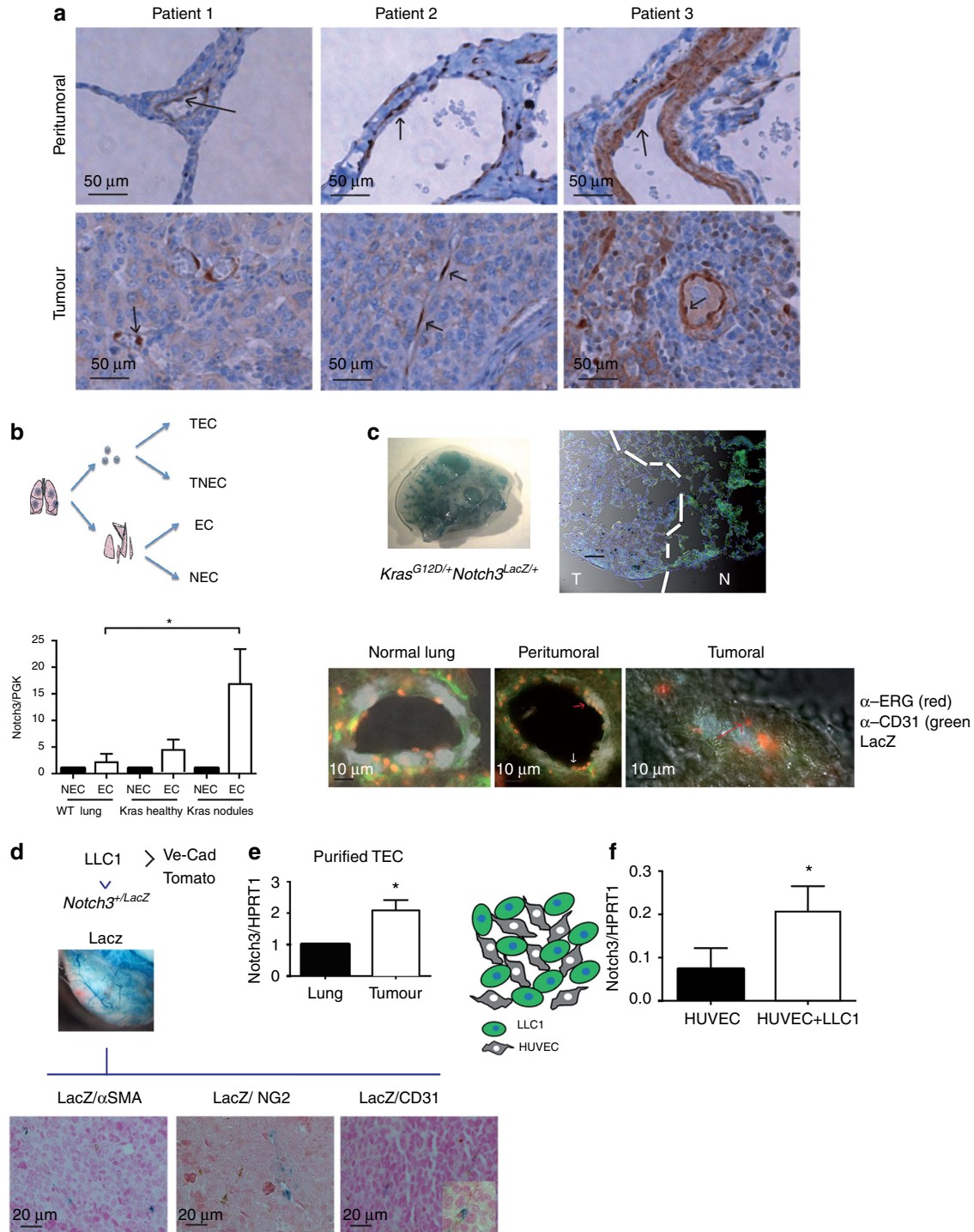

**Figure 1 | Notch3 is aberrantly expressed in tumour endothelial cells.** (**a**) Notch3 immunohistochemistry on peritumoral and tumour part of three sections from non-small cell lung cancers patients. (**b**) Quantitative RT-PCR was performed to measure Notch3 mRNA expression in endothelial cell enriched fraction (EC, CD31-expressing purified cell population) or non endothelial cell (NEC) purified from lung dissected from wild-type mice, the healthy part of tumour-bearing lung dissected from Kras mice or from the nodules dissected from the lung of Kras mice ($n = 6$ WT lungs, $n = 5$ Kras lungs, mean ± s.e.m., ordinary one-way ANOVA, multiple comparisons). (**c,d**) β-galactosidase staining was performed on lungs or LLC1 tumour whole mount from $Kras^{G12D/+}$ (**c**) or WT mice (**d**) mice before inclusion in paraffin and immunohistochemistry staining for ERG, CD31, SMA, NG2 as indicated. (**e**) Quantitative RT–PCR was performed to measure Notch3 mRNA expression in endothelial cell FACS-sorted from lung or tumour dissected from Cdh5:Cre^ERT2^xTomato (Ve-Cad Tomato) mice ($n = 3$ tumours, mean ± s.e.m., unpaired $t$-test). (**f**) HUVEC cells were co-cultured for 48 h with LLC1 cells stably expressing GFP before being FACS sorted. DAPI (alive cells) GFP negative (HUVEC) cells were used to prepare mRNA and Notch3 expression was measured by quantitative RT–PCR ($n = 3$ independent experiments, paired ratio $t$-test).

staining was reported to be strongly specific for EC[28]. However, in the peritumoral part, we observed ERG staining in LacZ positive cells (red arrow) (Fig. 1c). We next injected LLC1 syngeneic lung cancers cells in $Notch3^{+/LacZ}$ to assess expression of Notch3 in subcutaneous graft. We confirmed that Notch3 is expressed in the vasculature of the grafted tumours and as observed in the tumour nodules from the $Kras^{+/G12D}$ mice, we observed an aberrant expression of Notch3 in tumour-associated EC (Fig. 1d,e): Notch3 co-localizes with CD31 (endothelial cell marker platelet/endothelial cell adhesion molecule, PECAM-1) but not with mural cell markers αSMA or NG2 (Fig. 1d). We further confirmed the up-regulation of Notch3 mRNA in purified tumour-associated endothelial cells from subcutanous injected LLC1 in Cdh5:Cre$^{ERT2}$xTomato mice allowing FACS sorting of tumour-associated endothelial cells (Fig. 1e). The LLC1 model thus provides a good model to study the functional impact of Notch3 aberrant expression on tumour vasculature. Furthermore, co-culturing LLC1 cells with HUVEC was sufficient to induce an upregulation of Notch3 in the endothelial cells, showing that the epithelial cancer cells are sufficient to induce Notch3 expression in endothelial cells (Fig. 1f).

**Stroma specific Notch3 silencing promotes tumour angiogenesis.** We next assessed the role of this aberrant expression of Notch3 in tumour vasculature by establishing a model in which Notch3 is silenced only in the stroma but not in the tumour cells. As we started with observations in human lung carcinomas, we chose the murine lung carcinoma LLC1 syngeneic grafts in wild-type and in $Notch3^{LacZ/LacZ}$ mice. As shown in Fig. 2a, the absence of stromal Notch3 was associated with an increase of tumour growth. This suggests that the endothelial expression of Notch3 limits tumour angiogenesis. This observation was also true in another model of syngeneic graft, the E0771 mammary gland tumour model, although to a lesser extend (Supplementary Fig. 3a). In line with a role of Notch3 in tumour associated endothelial cells, we observed an increase in CD31 and DLL4 expression in tumours from $Notch3^{LacZ/LacZ}$ mice (Fig. 1b and supplementary Fig. 3b), but no change in αSMA or PDGFRβ (Beta-type platelet-derived growth factor receptor), two pericyte markers (Fig. 2b). As Notch3 has been reported to be expressed in certain immune cells[29], we looked for the expression of CD11b and CD45 that remained unchanged (Fig. 2b). We next looked at the vascularization of tumours grown in the absence of stromal Notch3 expression. CD31 staining of tumours grown in the wild-type mice or in the $Notch3^{LacZ/LacZ}$ mice showed an increased vascularization in the latter (Fig. 2c). Furthermore expression of αSMA in these tumours was unchanged (Fig. 2c). This suggests that the aberrant expression of Notch3 in tumour endothelial cells could limit tumour angiogenesis whereas the absence of Notch3 in vascular smooth muscle cells has no effect. Furthermore this effect seems to be independent of the normal role of Notch3 in smooth-muscle cells.

**Notch3 behaves as a dependence receptor.** To understand how the absence of Notch3 would impact the tumour vascularization, we studied *in vitro* human umbilical vein endothelial cells (HUVEC). As described previously[9], these cells express a low level of Notch3 which is almost entirely cleaved into N3ICD as treatment with DAPT completely abolished the presence of a 75 kDa band recognized by a C-terminal antibody (Supplementary Fig. 4a). We then asked what would be the consequence of an upregulation of Notch3 in these cells that would mimic the aberrant expression of Notch3 observed in lung cancers-associated endothelial cells. We first used electroporation in HUVEC cells (with 80% electroporation efficiency

(Supplementary Fig. 4b)). As shown in Fig. 3, Notch3 forced expression in HUVECs triggered cell death as evidenced by an increase of the sub-G1 cell population (Fig. 3a) and Annexin-V-positive cell population (Fig. 3b). This cell death is probably, at least in part, apoptosis as it is inhibited by general caspase inhibitor z-VAD-fmk (Fig. 3c). At this stage we cannot however exclude that Notch3-induced cell death is not only apoptosis as inhibition of cell death by caspase inhibitors is not complete. Interestingly, N3ICD did not induce cell death (Supplementary Fig. 4c). Furthermore, although Notch3 level is low in HUVEC under normal condition, knocking-down Notch3 in a setting of network formation in matrigel was sufficient to inhibit significantly apoptosis during network regression (Supplementary Fig. 4d). We further used the S1-Cter Notch3 construct (a truncated version of Notch3 (S1-Cter Notch3; aa1573 (furin cleavage site) to the C terminus, Supplementary Fig. 4e) as it mimics the absence of ligands and also helps bypassing the possible effect of varying levels of ligand expression in different cellular models. S1-Cter Notch3 induced very low Notch transcriptional activity in comparison to N3ICD (Supplementary Fig. 4f). Whereas S1-CterN3 expression induced caspase-3 cleavage (Supplementary Fig. 4g), electroporation of N3ICD, or of CBF1-VP16 (which both activates canonical Notch signalling in HUVEC (Supplementary Fig. 4f)), or of DNMAML (Dominant negative Mastermind-like, which inhibits endogenous Notch signalling (Supplementary Fig. 4f)) had no effect on induction of cell death (Supplementary Fig. 4g,h), suggesting that canonical Notch signalling is not involved in this process. Of interest, S1-Cter Notch3 mutant, that fails to interact with the CBF1 transcription factor (S1-Cter WFP-LAA), is still able to induce caspase-3 cleavage (Supplementary Fig. 4h) supporting the view that the canonical Notch3 signalling pathway is not involved here. Such ability of a transmembrane receptor to trigger apoptosis in a setting of absence of ligand, recalls the behaviour of dependence receptors[30]. Such receptors that include the netrin-1 receptors DCC and UNC5H (ref. 31) or the Hedgehog receptors Ptc and CDON[21,22] share the ability to actively transduce a death signal in settings of ligand limitation, thus creating a state of cellular dependence to the presence of ligand for cell survival. Most of these dependence receptors share the trait of being cleaved by caspase[30]. We thus looked whether Notch3 could similarly be cleaved by caspases. Expression of S1-Cter Notch3 or an S2-Cter Notch3 (aa1631 to the C terminus) in HEK293T cells allows the identification of a 60–65 kD N-terminal fragment and a lower size 25–30 kD Notch3 C-terminal reactive fragment (Supplementary Fig. 4i,j). These fragments were no longer detected upon incubation with z-VAD-fmk and more specifically with initiator caspase inhibitors IETD-fmk and LEHD-fmk, supporting the view that a Notch3 fragment is released upon a caspase-like dependent cleavage (Supplementary Fig. 4i). To map more precisely the caspase-cleavage site in Notch3, systematic mutations of aspartic acid residues were performed. The specific mutations of the aspartic acid residues 2104 and 2107 into asparagine residues (D2104N-D2107N) fully blocked the detection of the Notch3 fragment (Supplementary Fig. 4j) without affecting canonical Notch signalling (Supplementary Fig. 4k). Thus, Notch3 is cleaved by a caspase-like protease at DSLD (2104–2107). Interestingly, this cleavage site is not present in other Notch receptors but is conserved in Notch3 receptors (Supplementary Fig. 4l). Therefore, expression of Notch3 *in vitro* induces cell death of EC, and Notch3 is cleaved by caspase-like proteases. Another frequent characteristic of dependence receptors is their ability to recruit and activate the initiator caspase-9 (ref. 30). We first observed that caspase-9 might be required for Notch-3-induced cell death as treatment with z-LEHD-fmk significantly inhibited cell death induced by Notch3 over-expression (Fig. 3c). We further confirm the importance of caspase-9 by analysing Notch3-induced

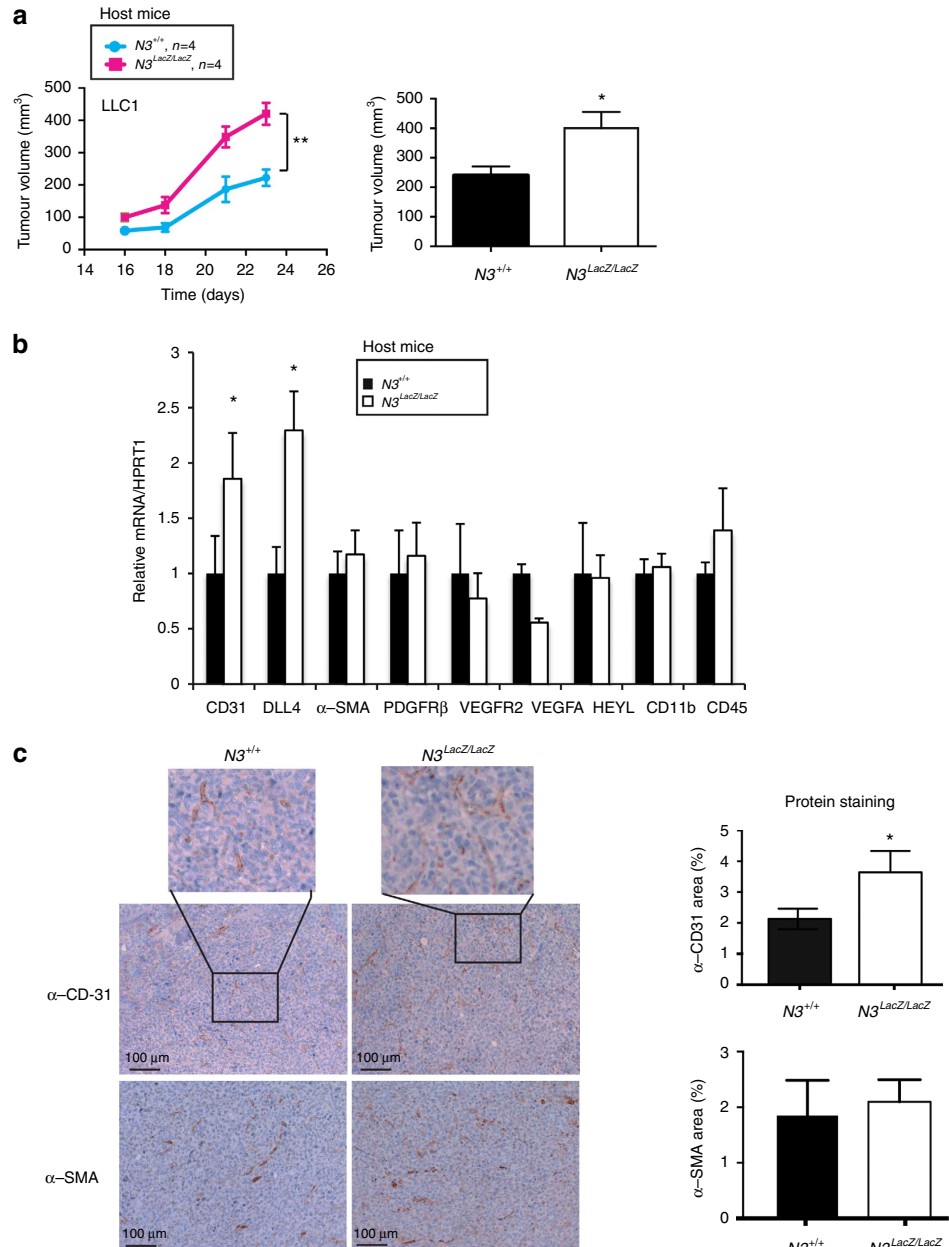

**Figure 2 | Notch3 limits tumour growth and vascularization *in vivo*.** (**a**) $5 \times 10^5$ LLC1 cells were implanted into the left flank of wild-type C57Bl/6 mice
($N3^{+/+}$, $n = 4$) or of Notch3 LacZ homozygous Knock-in C57Bl/6 littermates ($N3^{LacZ/LacZ}$, $n = 4$). Tumour growth was monitored from day 16 until day 24
when mice were sacrificed. Two-way ANOVA was performed to assess time and genotype effect on tumour growth (Interaction: $P = 0,013$;
Time: $P < 0,0001$; Genotype: $P = 0,0015$). (**b**) mRNA was extracted from tumours dissected after 14 days of growth from wild-type C57Bl/6 mice
($N3^{+/+}$, $n = 6$) or Notch3 mutant mice $N3^{LacZ/LacZ}$ C57Bl/6 littermates ($N3^{LacZ/LacZ}$, $n = 5$). Quantitative RT–PCR was performed to measure CD31, DLL4,
PDGFRβ, α–SMA, VEGFR2, VEGFA, CD11b and CD45 expression (means ± s.d., unpaired $t$-test was applied). (**c**) Immunohistochemistry for CD31 and
α–SMA was performed on tumours dissected from wild-type mice ($N3^{+/+}$, $n = 15$) or Notch3 mutant ($Notch3^{LacZ/LacZ}$) mice littermates ($n = 9$) on day 14.
Images are representative of four different sections from each tumour. Quantification was done using ImageJ angiogenesis plug-in on four different images
from each tumour (mean ± s.e.m. for quantification, unpaired $t$-test).

cell death upon silencing of caspase-9. As shown in Fig. 3d, silencing of caspase-9 strongly inhibits cell death induced by Notch3 (Fig. 3d). We then asked whether Notch3 could interact with caspase-9. Interestingly, we observed that S1-Cter Notch3, but not S1-Cter Notch1 or S1-Cter Notch2, was able to interact with caspase-9 when both Notch proteins and caspase-9 were ectopically expressed (Fig. 3e). We confirmed the interaction between Notch3 and caspase-9 by immunoprecipitation of endogenous caspase-9 (Fig. 3f). Interestingly, N3ICD did not interact with caspase-9

under the same condition, suggesting that the interaction with caspase-9 needs the anchorage of Notch3 to the membrane. We also performed Proximity Ligation Assay (PLA) with endogenous caspase-9 upon Notch3 overexpression in HEK293T cells. We observed a clear interaction between Notch3 and caspase-9 whereas no interaction was observed with caspase-8 (Fig. 3g). Moreover to explore whether the recruited caspase-9 could be activated, we performed caspase-9 activity assessment on Notch3 pull-down. As shown in Fig. 3h, Notch3, but not N3ICD, was pulling down

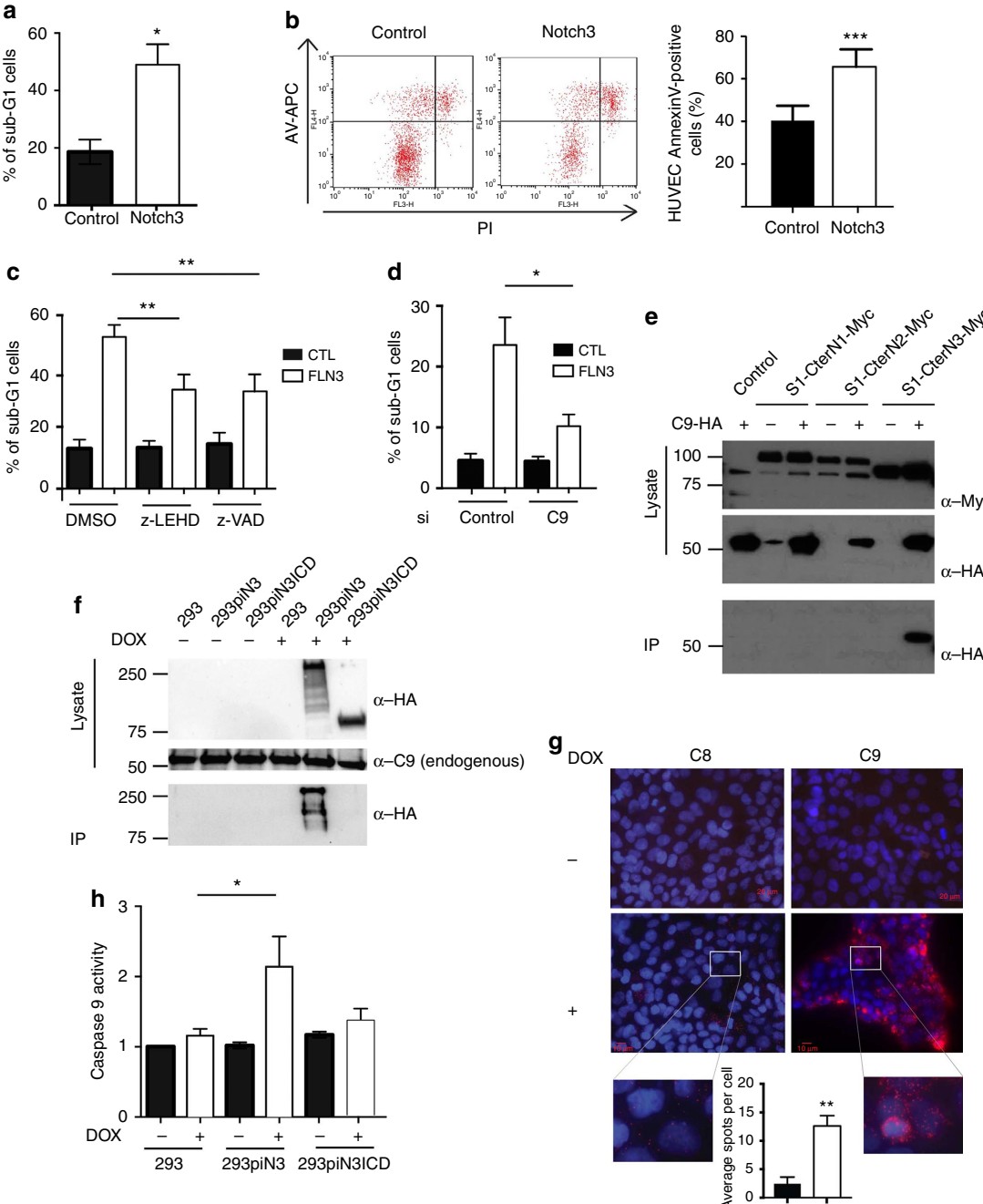

**Figure 3 | Notch3 induces endothelial cell death *in vitro*.** (**a**) Sub-G1 analysis of HUVEC electroporated with a control plasmid (Control), or a plasmid expressing the full-length version of Notch3 for 24 h. Quantification was made for three independent experiments and a paired-ratio *t*-test was applied. (**b**) AnnexinV/Propidium iodide was performed on HUVEC cells electroporated with a control plasmid or a plasmid expressing Notch3. Quantification was made on three independent experiments and a paired ratio student *t*-test was applied. (**c**) Sub-G1 quantification was made after electroporation of HUVEC cells with a control plasmid or a plasmid expressing Notch3 after 48 h of treatment with DMSO or z-LEHD-fmk or z-VAD-fmk pan-caspase inhibitors. Quantification was made on three independent experiments and a paired ratio student *t*-test was applied. (**d**) HUVEC were electroporated with a control siRNA (control) or a siRNA targeting caspase-9 (C9). 48 h later, cells were electroporated with a control plasmid or a plasmid expressing Notch3 and 24 h later, sub-G1 analysis was made. Quantification was made on three independent experiments and a paired ratio student *t*-test was applied. (**e**) Immunoprecipitation of Myc-tagged S1-Cter Notch1 (S1-CterN1), S1-Cter Notch2 (S1-CterN2) and S1-Cter Notch3 (S1-CterN3) constructs in HEK293t cells together with a control plasmid (−) or a plasmid expressing an HA-tagged dominant-negative version of caspase-9 (C9). (**f**) Lysates form HEK293t cells carrying Doxycycline (DOX)-inducible HA-tagged Notch3 (piN3) or DOX-inducible HA-tagged N3ICD (piN3ICD) plasmids were immoprecipitated for endogenous caspase-9 and western blot were done to analyse Notch3 (HA) or caspase-9 in total lysates or IP. (**g**) Proximity Ligation Assay was performed for endogenous caspase-9 or endogenous caspase-8 and doxycycline-induced Notch3 in HEK293 cells. Quantification was made on five images containing 100–150 cells each. (**h**) Caspase-9 activity was measured from Notch3 immunoprecipitated lysates from HEK293t cells carrying inducible Notch3 (piN3) or inducible N3ICD (piN3ICD) plasmids.

caspase-9 activity, supporting the view that Notch3 could trigger cell death similarly to other dependence receptors.

These observations prompted us to further investigate whether Notch3 could be a dependence receptor for tumour EC aberrantly expressing Notch3. As a dependence receptor, it is expected that Notch3 ligand blocks Notch3 induced endothelial cell death. As Jag-1, a Notch3 ligand, has been shown to be associated with increased tumour angiogenesis[32], we looked for the effect of Jag-1 expression on Notch3-induced cell death in tumour. For this purpose, HUVEC were co-cultured with two lung carcinoma cell lines expressing low or high levels of Jag-1, murine LLC1 cells and human H358 cells, respectively (Supplementary Fig. 5a). We observed that over-expression of Jag-1 in LLC1 cells reduced endothelial cell apoptosis and therefore induced stabilization of the endothelial network (Fig. 4a,b). Conversely, silencing Jag-1 in H358 cells led to an increase in endothelial cell apoptosis and earlier destabilization of the endothelial network (Fig. 4a,b). To confirm *in vivo* that tumour-derived expression of Jag-1 could increase angiogenesis, we established graft of LLC1 overexpressing Jag-1. As shown in Fig. 4c, overexpression of Jag-1 in LLC1 cells induced a dramatic increase in angiogenic markers CD31 as seen both on mRNA level and on protein staining by immunohistochemistry (Fig. 4c). To go further and prove that Notch3 behaves as a dependence receptor, we then over-expressed Notch3 in HUVEC cells and co-cultured them with LLC1 cells expressing or not high level of Jag-1. Overexpression of Jag-1 in LLC1 cells rescued the HUVEC death induced by Notch3 (Fig. 4d). We also showed that in co-culture conditions, neither N3ICD nor DNMAML was able to induce cell death (Supplementary Fig. 5b). This further supports the view that Notch3 induces endothelial cell death independently of Notch canonical signalling pathway, and that the expression of Jag-1 by cancer cells can cell non-autonomously rescue endothelial cell death. Jag-1 is also frequently over-expressed in epithelial cancer cells[33,34]. We observed that Jag-1 was overexpressed in a fraction of human lung cancers using the GSE7670 data set (Fig. 4e) and confirmed Jag-1 over-expression in human clear cell renal cell carcinomas (Supplementary Fig. 5c). Of interest, Jag-1 expression was only poorly correlated with HES1, HEY1 and HEYL Notch target genes expression in this data set as well as in the GSE10245 dataset (Supplementary Fig. 5d). This observation supports the hypothesis that Jag-1 could have a different role in the tumour than activating Notch canonical signalling. As Jag-1 was shown to have a paradoxical pro-angiogenic role regarding Notch activation[35], we compared the expression of Jag-1 with the expression of CD31 among tumours that over-express Jag-1 in the GSE7670 data set. In these patients, we observed a strong correlation with CD31 expression (Fig. 4e). We observed the same correlation in clear cell renal cell carcinoma (Supplementary Fig. 5c). By carrying out non-supervised clustering using the GSE10245 dataset, we observed a population in which Jag-1 and CD31 clustered together whereas Jag-1 did not cluster with Notch target genes (Supplementary Fig. 5e). In this population we observed a strong correlation between Jag-1 and CD31 but not with Notch target genes (Supplementary Fig. 5f). Taken together these data support the view that Notch3 behaves as a dependence receptor in endothelial cells and that Jagged-1 expression in tumour may act as a pro-angiogenic mechanism by limiting Notch3 induced apoptosis in endothelial tumour cells.

**Notch3 is required for γ-secretase-induced tumour regression**. We then hypothesized that γ-secretase inhibitors, by blocking the N3ICD formation may mimic the absence of Notch3 ligand and thus induce Notch3-dependent tumour-associated endothelial cell death. The general view for the mode of action of γ-secretase inhibitors as anticancer agents is the inhibition of cancer cell proliferation. However, γ-secretase inhibitors treatments have been paradoxically associated with decreased angiogenesis[12,13] and endothelial cell death[11] as opposed to anti-Dll4 antibody treatment which induces increase in non-productive angiogenesis[14]. We first observed that, *in vitro,* DAPT treatment induced HUVEC cell death (Fig. 5a). Of interest, this cell death was rescued by silencing Notch3 (Fig. 5b) but not by silencing Notch1 or Notch2 which had no effect on DAPT induced cell death (Supplementary Fig. 6a). We confirmed that tumour-associated endothelial cells were sensitive to DAPT treatment by purifying endothelial cells from tumours of $Kras^{G12D/+}$ mice (Fig. 5c). Further confirming the role of Notch3 in DAPT-induced cell death, we also showed that tumour-associated endothelial cells of tumours purified from $Notch3^{LacZ/LacZ};Kras^{G12D/+}$ mice were not sensitive to DAPT treatment (Fig. 5c). We then asked whether this effect could also be seen *in vivo*. We therefore treated wild-type mice bearing LLC1 tumours with DAPT. Whereas DAPT had no effect on LLC1 cells *in vitro* (Supplementary Fig. 6b), DAPT treatment in wild-type mice was associated with tumour growth inhibition (Supplementary Fig. 6c). As described by others[11–13], this reduction was associated with a regression of the tumour vasculature as seen here by a decrease of CD31 staining and of the collagen IV/CD31 co-staining, which shows a regression of pre-existing vessels (Supplementary Fig. 6d). This tumour growth inhibition induced by DAPT treatment would classically be attributed to canonical Notch signalling inhibition. However, we report here that the tumour growth inhibitory effect of DAPT treatment was no longer observed in *Notch3* mutant mice (Fig. 5d). Because Notch3 is only silenced in stromal cells, this phenotypic rescue can only point out an effect of DAPT treatment on the stroma and cannot be easily explained by a difference in the canonical pathway (that is, if DAPT is inhibiting tumour angiogenesis by blocking the canonical pathway induced by Notch receptors, knocking down Notch3 should only add more tumour angiogenesis inhibition). In line with this, HeyL mRNA expression was not affected in both wild type and *Notch3* mutant mice in presence of DAPT (Supplementary Fig. 6e). We further purified tumour-associated endothelial cells and treated these cells with DAPT. In this setting, we saw no significant downregulation of Notch target genes HeyL, Hes1 and Hey1 (Supplementary Fig. 6f). In agreement with an effect on vasculature, we observed an increase in necrotic area in wild-type mice treated with DAPT but not in *Notch3* mutant mice (Fig. 5d). Confirming Cook *et al.*[11] data obtained in a different model, we observed increased endothelial cell death in wild-type mice treated with DAPT (Fig. 5e). In contrast, no effect was seen in *Notch3* mutant mice (Fig. 5e). This indicates that the apoptotic pathway mediated by Notch3 accounts, at least in part, for the regression of the tumour vasculature following DAPT treatment.

## Discussion

We uncovered here an unexpected function of Notch3 expression in tumour vasculature. Whereas Notch3 is normally expressed in smooth-muscle cells surrounding large vessels, we observed that Notch3 was upregulated in tumour endothelial cells. We have observed this ectopic expression in human lung cancer samples regardless of the expression of Notch3 in the cancer cells (Fig. 1; Supplementary Fig. 1). This expression was also observed in mice predisposed to develop lung cancers ($Kras^{+/G12D}$) as well as in lung cancer cells grafted subcutaneously (Fig. 1). These results are in line with the transcriptomic analysis data obtained by others[19]. Interestingly, although Notch3 has been

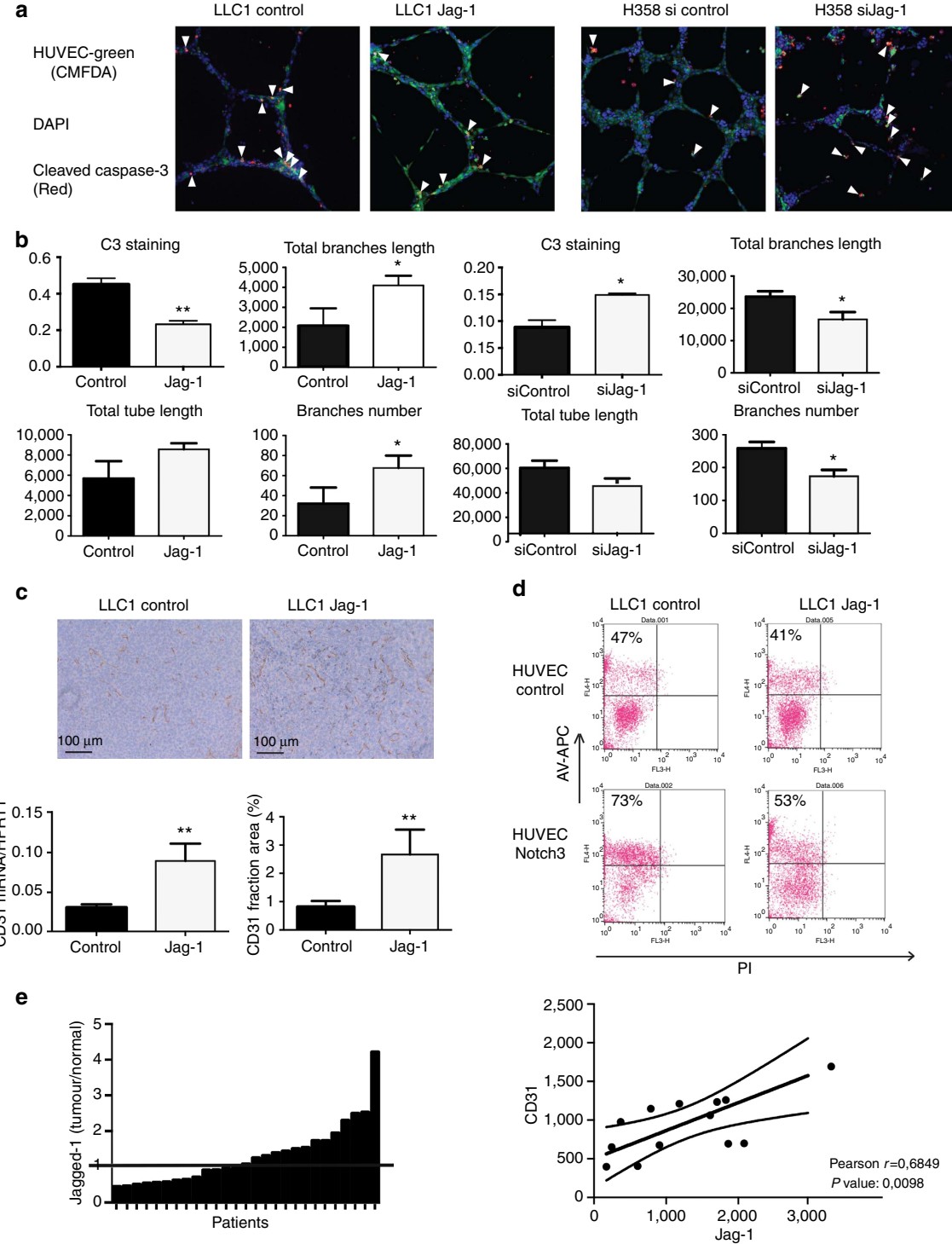

**Figure 4 | Jag-1 rescues Notch3-induced endothelial cell death. (a)** HUVEC were stained with the CellTRacker green CMFDA. Cleaved-caspase-3 staining of HUVEC co-cultured with 1) LLC1 expressing or not Jag-1 (LLC1-Jag-1 and LLC1-Control, respectively) or 2) H358 cells transfected with a siRNA control (sicontrol) or a siRNA targeting Jag-1 (siJag-1). Co-cultures were maintained in matrigel for 9 h before being fixed and stained. Images are representative of three independent experiments, each performed in triplicate. **(b)** Quantification of HUVEC networks presented in a ($n = 3$, means ± s.d., $t$-test was applied). Jag-1 expression was verified on western blot before the cells were added to HUVEC in Matrigel. **(c)** Immunohistochemistry staining and quantification of CD31 on tumour section from tumours obtained from LLC1 cells or LLC1 cells overexpressing Jag-1 ($n = 5$ tumours, quantification on 4 images/tumours, unpaired $t$-test). **(d)** Co-culture of HUVEC electroporated (Notch3) or not (Control) with Notch3 and LLC1 cells transfected (LLC1-Jag-1) or not (LLC1 control) with Jag-1 were stained with Annexin V APC and studied by flow cytometry. HUVEC were gated (M1) according to FL1 staining (cellTracker green CMFDA), and Annexin V (FL4)-positive cells were quantified among HUVEC. Number of Annexin V positive HUVEC cells is specified in each condition. **(e)** Tumour/normal tissue ratio of Jag-1 mRNA in patients with non-small-cell lung adenocarcinomas from the GSE7670 data set and Correlation between Jag-1 and Pecam-1 expression in patients overexpressing Jag-1 in the GSE7670 data set.

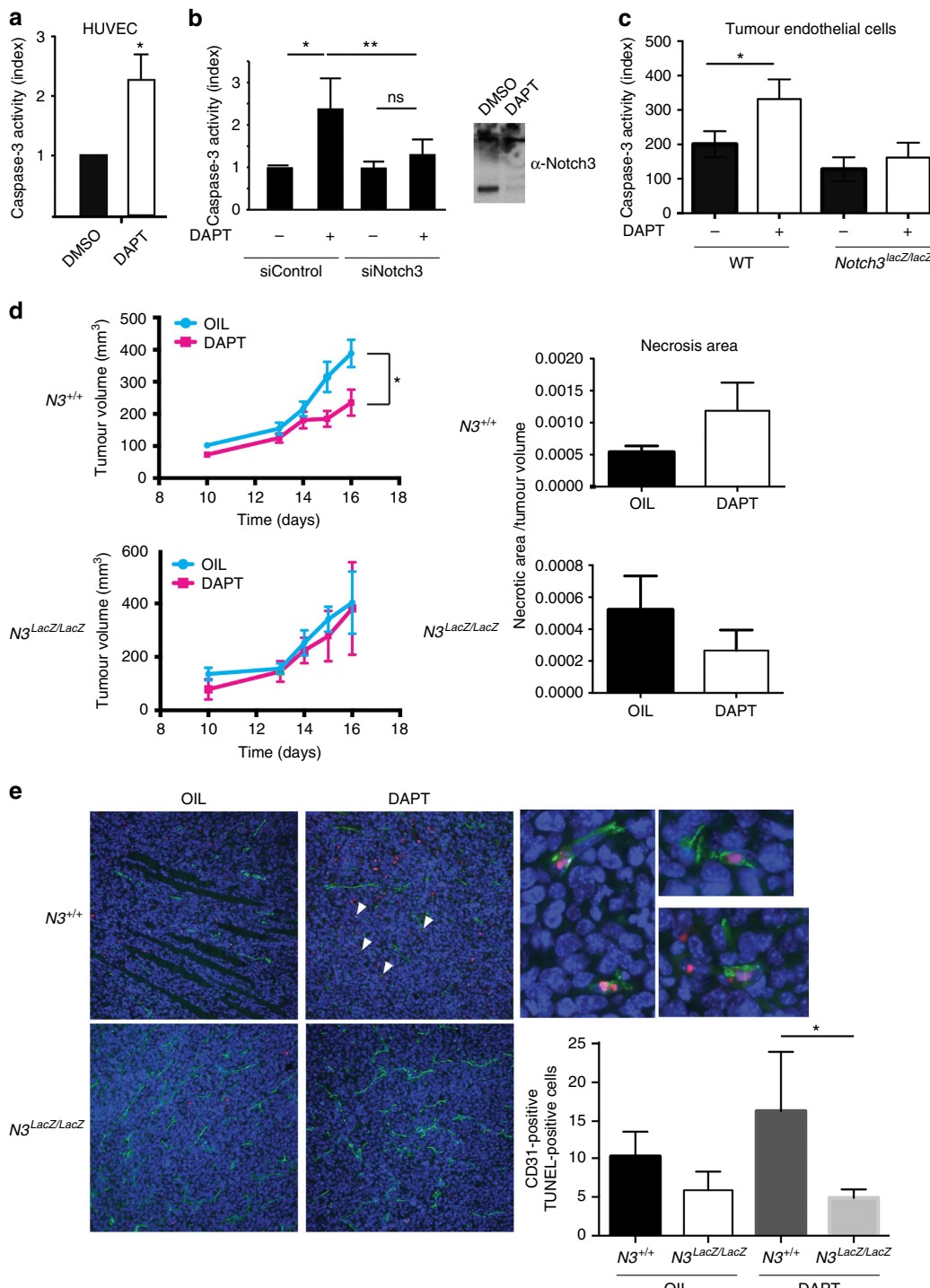

**Figure 5 | Notch3 is required for γ-secretase-induced tumour regression.** (**a**) Caspase-3 activity determined in HUVEC treated with DAPT (4 μM) for 48 h (n = 7, means ± s.d., paired t-test was applied). (**b**) Caspase-3 activity in HUVEC electroporated with siRNA control (sicontrol) or a siRNA targeting *Notch3* (siNotch3) and treated ( + ) or not ( − ) with DAPT (4 μM) (n = 7, mean + / − s.d., paired t-test was applied). Effect of DAPT treatment on Notch3 cleavage was monitored by western blot. (**c**) Caspase-3 activity was determined in lysates from purified tumour-associated endothelial cells from tumours dissected from *Kras*^G12D/ + ;*Notch3*^+/+ (WT) or *Kras*^G12D/ + ;*Notch3*^LacZ/LacZ (*Notch3*^LacZ/LacZ) mice and treated *in vitro* (DAPT) or not with 4 μM of DAPT. (**d**) 5 × 10^5 LLC1 cells were implanted into the left flank of wild-type C57Bl/6 mice or of *Notch3*^LacZ/LacZ C57Bl/6 littermates, and injected intraperitoneally with 10 μl g^−1 of ethanol-corn oil (1/9) (*N3*^+/+ OIL, n = 10, *N3*^LacZ/LacZ OIL, n = 4) or 10 μl g^−4 of 1 mg ml^−l DAPT diluted in ethanol-corn oil (1/9; *N3*^+/+ DAPT, n = 8; *N3*^LacZ/LacZ DAPT, n = 4) on days 12,13,14 and 15. Mice were killed on day 16 (two-way ANOVA was performed to test for significance). Necrosis area was quantified automatically on whole sections with HistoLab software settings parameters on hematoxylin staining intensity. (**e**) Immunofluorescence and TUNEL staining were performed on LLC1 tumour sections from wild-type mice or *Notch3*^LacZ/LacZ littermates treated (DAPT, *N3*^+/+ DAPT, n = 5; *N3*^LacZ/LacZ DAPT, n = 3) or not (OIL, *N3*^+/+ OIL, n = 5; *N3*^LacZ/LacZ, n = 3) with DAPT as previously described. For each tumour, the entire section was imaged (8–12 images per tumour), and the number of CD31-positive TUNEL-positive cells was quantified. Number of TUNEL-positive cell was then normalized on CD31 area that was quantified on each image using the ImageJ angiogenesis plugin (mean ± s.e.m., unpaired t-test).

shown to be involved in different pathological settings affecting the vasculature, its role in tumour vasculature has never been addressed. Here we showed that Notch3 behaves as a dependence receptor, regulating tumour angiogenesis. As for the other dependence receptors, this new function is independent on the canonical Notch signalling pathway. Indeed, activation or inhibition of the canonical Notch signalling by expression of a dominant version of Mastermind-like or a constitutive active CBF-1 do not induce cell death as does Notch3. Furthermore, mutating the residues necessary for the interaction between N3ICD and CBF-1 did not abrogate the ability of Notch3 to induce cell death. Interestingly, we showed here that Notch3 was the only receptor of the Notch family to present this function. This is also described for other dependence receptors: for example TrkA and TrkC behave as dependence receptors whereas TrkB does not[36,37]. Interestingly, Notch3 has been shown to arose from the second duplication of Notch1 (ref. 38). As hypothesized for other dependence receptors, the dependence receptor function of Notch3 is thus probably a late acquisition during evolution. In line with this, the caspase cleavage site, present only in Notch3, has likely appeared during Notch3 differentiation after duplication from Notch1. Notch4 has been proposed to derive from Notch3 (ref. 39), however, this has been questioned more recently[38] and due to its rapid evolution, it is not clear from which Notch gene it actually derives.

This function of Notch3 appears in a context in which Notch3 is aberrantly expressed in the pathological tumour vascularization where Notch3 limits tumour angiogenesis through an unexpected pro-apoptotic activity. Of note, tumour associated endothelial cells have been described to have an aberrant expression of DR5 which render them more susceptible to apoptosis induced by TRAIL[40]. It would be of interest to study whether this function is conserved in other pathological situations where Notch3 is aberrantly expressed in non-endothelial cells, for example in cancer cells in which Notch3 and its ligands have been shown to be expressed.

We also observed that Notch3 was, at least in part, responsible for the anti-angiogenic effect of γ-secretase inhibitors described by others[11]. Indeed DAPT treatment induced a reduced vascularization associated with a reduced tumour growth. Importantly, this effect of DAPT was not due to inhibition of the canonical Notch signalling pathway as the effect of DAPT could be reversed by deletion of Notch3. If the effect of DAPT was a consequence of inhibition of Notch signalling, Notch3 deletion should either not have any effect or exasperate the effect of DAPT.

Furthermore, inhibition of the canonical Notch pathway, would lead to a hypersprouting of endothelial cells as observed upon anti-Dll4 treatment which could be associated with decreased growth but not decrease vasculariztion. In contrast, Notch3-induced apoptosis in tumour-associated endothelial cells following DAPT treatment could explain at least partly the anti-angiogenic effect followed by tumour growth inhibition. In Notch3 mutant setting, DAPT cannot trigger Notch3-induced apoptosis and thus angiogenic effect. It may thus be of interest to take this unexpected function of Notch3 into account when evaluating the anti-tumour efficacy of γ-secretase inhibitors. This function of Notch3 is not in contradiction with the well-described oncogenic canonical Notch3 signalling in epithelial cells[26,41]. In fact, as other dependence receptor, the availability of ligands would impact on the role of Notch3. We showed here that Jag-1 expression by cancer cells was important to limit the dependence receptor function of Notch3. Furthermore, the function we describe here in tumour angiogenesis could account for some paradoxical observations regarding Notch3. In fact, while it may play a role in the epithelial tumour cells as an oncogene through

its canonical signalling, it may also represent a constraint for tumour progression by acting as a cellular sentinel for endothelial cell death. The Notch3 receptor may therefore act as a regulator of tumour angiogenesis depending on the context such as the heterotypic interactions between the tumour and the stroma or the availability of the ligands in the tumours. Jag-1 has been shown to be very important in signalling from the endothelium to the cancer cells[42,43]. Together with the present data, it shows how reciprocal interactions between the tumour vasculature and the tumour are important. The data presented here also raise the question of targeting Notch to regulate tumour angiogenesis. We propose that targeting Jag-1 in tumour angiogenesis might therefore be an interesting approach and targeting more specifically the Notch3-Jag-1 interaction could be advantageous allowing targeting of both the canonical Notch signalling in epithelial cells and Notch3-induced apoptosis in endothelial cells.

## Methods

**Mice experiments.** Notch3 mutant mice have been characterized previously[16]. Cdh5:CreERT2 mouse line was generated by Ralph Adams at Cancer Research UK. Mice were constantly bred into C57Bl/6 mice and experiments have been conducted in agreement with the local ethic comity (CECCAPP, Comité d'Evaluation Commun au PBES, à AniCan, au laboratoire P4, à l'animalerie de transit de l'ENS, à l'animalerie de l'IGFL, au PRECI, à l'animalerie du Cours Albert Thomas, au CARRTEL INRA Thonon-les-Bains et à l'animalerie de transit de l'IBCP). LLC1 cells were purchased from ATCC and were tested for mycoplasmas and murine viruses (Murine essential panel, Charles River) before being implanted in mice. For sub-cutaneous engraftment, $5 \times 10^5$ LLC1 cells were implanted into the left flank of wild-type C57Bl/6 mice or Notch3$^{LacZ/LacZ}$ C57Bl/6 littermate. Standard variation was established in control experiment. Groups of 4–12 animals with homogenous tumour size were selected to obtain equal variance before treatment. No randomization method was applied. Tumour size was measured every day from day 10 when the tumours are palpable until day 14 or 21 by two different persons for each measure without knowing the genotype of animals. Animal showing prostration or obvious sign of suffering were excluded. Sub-cutaneous engraftment with E0771 cells was performed as described previously. Tumours were measured from day 14 to day 25. When the measures were too different, the point could be excluded. Measurement of the tumours was carried out without knowing the genotype of the animals. Mice were sacrificed before the end of the experiment if necessary according to animal care guidelines. For DAPT treatment, DAPT was diluted in Corn Oil/Ethanol (9/1) at 1 mg ml$^{-1}$. 10 µl g$^{-1}$ was injected intraperitoneally to reach a 10 mg kg$^{-1}$ concentration. Experiments were all conducted on male and female littermate of 4–7 weeks of age. Animals were treated according to their identification number (even = untreated; odd = treated, this was arbitrary chosen for each experiment). Tumour dissection, fixation, and immunochemistry analysis were performed simultaneously.

**Cell culture and cell transfection.** Human umbilical vein endothelial cells (HUVEC) were obtained from Promocell (Heidelberg, Germany) and maintained in Endothelial Cell Growth Medium 2, supplemented with Endothelial Cell Growth Medium 2 Supplement Mix and 1% penicillin/streptomycin. H358 and LLC-1 were obtained from the ATCC maintained in RPMI Medium 1640 $(1 \times)$ + GlutaMAX-I, supplemented with 10% fetal bovine serum (FBS) and 1% penicillin/streptomycin (PS) and in DMEM supplemented with 10% FBS and 1% PS, respectively. E0771 cells were obtained from our lab and culture in DMEM as described previously.

For electroporation, $1 \times 10^6$ HUVEC cells were collected by trypsinization and electroporated either with 10 nM siRNA (si Notch3, Sigma SASI_Hs01_00101286, Sigma SASI_hs01_00100441, si negative control Sigma #SIC001) or 5 µg DNA plasmids with Neon kit (Invitrogen). Twenty four hours later, transfection efficiency was verified by RT–quantitative PCR. LLC-1 or H358 cells were seeded at $0.25 \times 10^6$ cells in 6 wells plates one day before transfection. Transfections were performed with lipofectamine TM reagent (Invitrogen) following the manufacturer's instructions.

For caspase inhibitors treatment, HUVEC cells were pre-incubated 2 h with 5 µM caspase inhibitors (BioVision, Caspase-9 Inhibitor Z-LEHD-FMK, MerkMillipore, Z-VAD-FMK ) or DMSO for 2 h. Cells were then transfected with empty vector or Notch3 and incubated for 24 h with 5 µM caspase inhibitors or DMSO.

**Endothelial cells purification.** Lung from 16 weeks-old KrasG12D mice were dissected and tumour nodules extracted under a binocular before being digested in 1 mg/ml collagenase Type 1 (Invitrogen) for 1 h. Cell suspension was then incubated with magnetic beads (Dynabeads Sheep Anti-Rat IgG, Invitrogen) incubated overnight with CD31 antibody (clone MEC13.3, Pharmingen).

**β-galactosidase staining.** After dissection, organs from $Notch3^{LacZ/+}$ mice were fixed for 20 min before being washed three times in 0.2% NP-40, 0.01% NaDOC, 2 mM MgCl2 in PBS. Organs were then incubated for 1 h in 25 mM K3Fe(CN6), 25 mM K4Fe(CN6) Wash Buffer. Xgal reaction was then performed in 25 mM K3Fe(CN6), 25 mM K4Fe(CN6), 1 mg ml$^{-1}$ Xgal in Wash Buffer at 37 °C.

**Co-culture experiments.** HUVEC were incubated with a CellTracker Green CMFDA at 1.25 µg ml$^{-1}$ (Molecular probes, Life technologies, C7025) for 30 min. Afterwards, cells were washed two times with PBS. 60 µl of Basement Membrane Matrix (Matrigel, BD Bioscience) was added to a 96-wells plate, followed by 30 min incubation at 37 °C. HUVEC were collected by trypsinization and $15 \times 10^3$ HUVEC were added into each Matrigel coated well and incubated at 37 °C for two hours. LLC1 or H358 transfected one day before were then added to wells containing HUVEC. Each condition was carried out in triplicate. After 9 h of co-culture, cells were fixed with 4% paraformaldehyde for 30 min at room temperature and rinsed three times with PBS for 10 min at room temperature. Fixed cell culture plates were stocked at 4 °C.

**Immunofluorescence staining.** Immunofluorescence analysis was performed on tumours obtained from littermates. Fixation and staining were performed simultaneously. Paraffin embedded tissue samples were deparaffinized and heat-induced antigen retrieval was performed. Cells or tissue samples fixed with 4% PFA were permeabilized with PBS-0.2% Triton X-100 (TX-100) for 30 min at room temperature. Samples were then washed three times with PBS for 5 min. Samples were blocked in PBS with 4% bovine serum albumin (BSA), 2% normal donkey serum and 0.2% TX-100 for one hour. Samples were then washed three times with PBS for 5 min. Primary antibodies were diluted in the blocking solution: 1:500 dilution for anti-cleaved caspase 3 (Cell Signaling Asp175 5A1E Rabbit mAb), 1:100 dilution for anti-CD31 (Abcam, anti-CD31 ab28364), 1:100 dilution for anti-collagen IV (Abcam anti-collagen IV ab19808). Alexa-conjugated secondary antibodies (Alexa555-donkey anti rabbit, Alexa488-donkey anti rabbit) were used at 1:1,000 dilution. DAPI (0.5 µg ml$^{-1}$) was added at the end to stain nuclei. Images were acquired with Zeiss Axio fluorescence microscopy, NIS element AR 4.20.01 Nikon fluorescence microscopy and confocal microscopy.

**Proximity ligation assay.** HEK293T cells ($10^5$; stably selected to carry a Doxycycline-inducible plasmid for Notch3 or N3ICD) were seeded in lab-tek chamber (Thermo scientific, 4-well, 177399). After 24 h, cells were treated with 1 µg ml$^{-1}$ Doxycycline. After 24 h induction, cells were fixed with 4% PFA and PLA assay (Sigma, Duolink In Situ PLA) was performed according to the manufacturer's instructions. Anti-HA (sigma, H6908) and anti-caspase 9 (Santa Cruz, sc-73548) were used for primary antibodies.

**Cell death assay.** TUNEL assay: Detection of DNA fragmentation, a terminal deoxynucleotidyl transferase-mediated deoxyuridine triphosphate nick-end labeling (TUNEL) assay was performed by following the protocol of the TUNEL assay kit (Roche). Briefly, fixed cells or tissue samples were permeabilized with 0.2% TX-100 in PBS (30 min at room temperature), washed with PBS, incubated with 300 U ml$^{-1}$ TUNEL enzyme and 6 µmol l$^{-1}$ biotinylated deoxyuridine triphosphate (Roche Diagnostics, Mannheim, Germany). The extremities of the biotin coupled DNA were revealed by using Cy-3-coupled streptavidin (1:1,000 in PBS, Jackson Immunoresearch). The slides were washed with PBS, DAPI-stained, then washed with PBS and finally, mounted with Fluoromount G (SouthernBiothec). Images were acquired with Zeiss Axiovision fluorescence microscopy and NIS element AR 4.20.01 Nikon fluorescence microscopy. Caspase-3 activity assay: Cells were first harvested by scraping. Cell pellets were obtained by centrifugation at 4 °C and lysed. The caspase 3 activity assay was performed according to the manufacturer's instructions (Biovision caspase-3 colorimetric assay kit). Total protein concentrations were measured with the BCA assay kit using BSA as a standard (Pierce Biotechnology, Rockford, IL, USA). Absorbance readings were done on a TECAN infinite F500.

**Immunohistochemistry analysis.** Immunohistochemistry was performed on 4-µm-thick sections of formalin-fixed, paraffin-embedded and heat-treated (for antigen retrieval) tissues (DakoCytomation). Sections were stained with hematoxylin-eosin-safran and tumour endothelium was stained with an anti-CD31 antibody (1:50 Abcam, ab28364). Diaminobenzidine was used as chromogen. Images were acquired with a Zeiss Axiovert. The whole slide was scanned automatically with the Histolab 6.2.0 MICROVISION Instrument system. Necrotic and CD31 positive areas were quantified by Histolab 6.2.0 or ImageJ angiogenesis plugin. Staining of human sample was performed with the cell signaling anti-Notch3 antibody (D11B8).

**Co-culture images analysis.** Images of co-culture experiment were acquired with Zeiss Axiovision fluorescence microscopy. 8–12 images were acquired for each well at ×5 magnification. Images were analysed by Image J angiogenesis plugin (Gilles Carpentier, Faculté des Sciences et Technologie, Université Paris Est Creteil

Val-de-Marne, France). Briefly, channels were split automatically. On the GFP channel, total tube length, branches number and total length of branches were acquired by Analysis HUVEC Fluo program. Each condition was carried out in triplicate.

**Tumour section CD31 fluorescent images analysis.** Whole slides were scanned to acquired total images at ×10 magnification with a Zeiss Axiovision fluorescence microscopy. CD31 expression areas were analysis by Image J. Briefly, The image was converted into RGB stack format. CD31 staining was quantified by choosing threshold program and adjusting the threshold parameters. Once the threshold parameter was adjusted, it was always the same for each image and CD31 expression areas were measured automatically. TUNEL positive and CD31-positive cells were counted manually. For blinded quantification, images were organized in folder identified by letters by one person and quantified by another.

**Western blot.** Cells were lysed in SDS buffer (2% SDS, 150 mM NaCl, 50 mM Tris-HCl, pH 7.4). Cell extract was next centrifuged at 2,500$g$ for 5 min. Protein concentration was measured with the BCA assay kit (Pierce Biotechnology, Rockford, IL, USA) using Bovine Serum Albumine (BSA) as a standard according to manufacturer's instructions. The following antibodies were used: anti-Notch3 (1:1,000 dilution, Cell Signaling #2889), anti-Jagged1 (1:1,000 dilution, Santa Cruz C-20 sc-6011), anti-cleaved caspase 3 (1:1,000 dilution, Cell signaling Asp175 5A1E Rabbit mAb), anti-CBF-1 (1:1,000 dilution, Cell signaling #5313P), anti-GFP (1:2,000 dilution, Molecular probes A11122), anti-Myc (1:2,000 dilution, sigma, M5546) and anti-HA (1:5,000 dilution, Sigma H4908).

**Flow cytometry analysis of sub-G1 and Annexin V staining.** For the sub-G1 experiment, cells were harvested cells by trypsinization and counted. Cells were first washed once with PBS followed by the addition of cold 70% ethanol, vortexed, and then resuspended at 4 °C for 30 min. Samples were stocked at −20 °C. Ethanol was removed and the pellet washed with PBS. Staining solution (40 µg ml$^{-1}$ propidium iodide, 2 mg ml$^{-1}$ RNAse in PBS) was added.

For the Annexin V experiments, cells were collected by trypsinization and counted. Afterwards, 100 µl of a $1 \times 10^6$ cells solution was incubated with 5 µl Annexin V allophycocyanin conjugated (Life technologies, A35110) and 2 µl Propidium Iodide (Life technologies, V13242), for 15 min at room temperature.

For the CD31/CD105 staining, cells were detached in PBS/EDTA (5 mM) and $10^6$ cells were re-suspended for 20 min on ice in 100 µl PBS with anti-mouse-CD31-FITC (Ebioscience) and Anti-CD105 antibody ([MJ7/18] Phycoerythrin; Abcam) before analyse with the flow cytometer. Data acquisition and analysis were performed on a FACSCalibur using CellQuestPro software (BD Bioscience, San Jose, USA).

**Statistical analysis.** For tumour growth analysis, a two-way ANOVA was realized to test for effect of time and treatment. For analysis of in vitro experiments a normality test was realized when number of samples was sufficient (Shapiro-Wilk or KS Normality test). Similarity of variance was tested before application of any statistical test using graphpad. If samples followed a Gaussian distribution, a $t$-test was applied, either paired-ratio or unpaired depending on the experimental data. When samples did not pass the normality test, non-parametric test was applied (Mann-Whitney for unpaired samples and Wilcoxon signed-rank test for paired samples). $*P < 0,05$; $**P < 0,01$; $***P < 0,001$.

**Quantitative RT-PCR.** mRNA were extracted with the NucleoSpin RNA kit (Machery-Nagel) according to manufacturer's instructions. cDNA were generated with the iScript cDNA Synthesis kit (BIO-RAD) according to the manufacturer's instructions. Real-time quantitative RT-PCR was performed using a LightCycler 480 (Roche Applied Science) and the FastStart TaqMan Probe Master Mix (Roche Applied Science). Primer sequences are provided in Supplementary Table 1.

**Immunoprecipitation.** $5.10^6$ HEK293T cells (stably selected to carry a Doxycycline-inducible plasmid for Notch3 or N3ICD) were treated with 1 µg ml$^{-1}$ Doxycyline for 24 h. Cells were collected and lysed in lysis buffer (HEPES 50 mM, NaCl 150 mM, EDTA 5 mM, NP40 0.1%, PH7.6) for one hour at 4 °C and then sonicated. One millilitre of lysate was then incubated with 10 µl Anti-HA (Sigma, H6908-.5ML) over night at 4 °C. Hundred microlitre of protein A sepharose beads (Sigma, P3391-1G) were added into the lysate and incubated at 4 °C for one hour. Beads were then washed three times with lysis buffer at 4 °C. Beads were collected and incubated with caspase 9 assay kit (Promega, Caspase-Glo 9 Assay) for 30 min. Luminescence was measured by TECAN infinite F500.

**Data availability.** All data are available within the Article and Supplementary Files, or available from the authors upon request.

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

## Acknowledgements

We are grateful to Sylvia Fre and Spyros Artavanis-Tsakonas for providing tools and for the critical reading of the manuscript. We also thank Ralph Adams for the Cdh5:Cre$^{ERT2}$ mouse line. This work was supported by institutional grants from CNRS, University of Lyon, Centre Léon Bérard and from the Ligue Contre le Cancer, INCA, ANR, ERC and Fondation Bettencourt (P.M.). O.M. is recipient of a Chair of Excellence INSERM-UCBL1.

## Author contributions

S.L. and A.N. designed and conducted most experiments. S.B. helped in conducting experiments. N.R., B.G., B.D., N.G. and J.-G.D. helped in designing experiments and provided critical advices. P.S. selected human lung cancer samples. I.T. analysed the staining of human lung cancer samples. O.M. designed and conducted experiments, analysed the data and together with P.M. designed the research plan and wrote the manuscript.

## Additional information

**Competing interests:** The authors declare no competing financial interests.

