## [Peer Review File · Nature Communications]

Reviewers' comments:

Reviewer #1 (Remarks to the Author):

The reviewers consider the manuscript has been improved. We appreciate that the authors addressed most of the comments from the first revision. However there are still two major issues that we will like to see clarified if not experimentally at least in the text and discussion.

1. Related to the comment 5. The authors provided an experiment with caspase inhibitors to prove that the cell death observed is caspase dependent. How do the authors explain that the cell death is only partially inhibited either with Casp9 or pan-caspase inhibitors? Is the NOTCH3 cleavage cause or consequence of the cell death observed in vitro? The lack of detection of the endogenous cleavage makes difficult to understand and relate the effects shown in HUVECs with the molecular mechanism proposed in the 293 cell overexpressing model.

2. The authors' statement in response to comment 9 is not correct - For the knowledge of the authors NOTCH3 floxed mice have been reported in literature i.e.

<http://hyper.ahajournals.org/content/early/2016/06/13/HYPERTENSIO NAHA.116.07694.abstract>

<https://www.ncbi.nlm.nih.gov/pubmed/21536678>

Reviewer #2 (Remarks to the Author):

Here the authors suggest that Notch3 acts as a dependence receptor in endothelial cells and regulates tumour angiogenesis by regulating endothelial cell death via caspase-9 activation and apoptosis downstream.

1. Given that the authors are claiming that Notch3 associates with caspase-9 and promotes activation of the latter, I am surprised that they did not attempt to assess the effect of caspase-9 knockdown or deletion (via CRISPR) on cell death induced by Notch3. Moreover, the claim that Notch3 associates with caspase-9 is based on very little evidence that is not properly controlled.

2. How Notch3 promotes apoptosis is very obscure in the paper. The authors claim that Notch3 acts as a direct regulator of caspase-9 activation, but this is based on the most superficial of data. This is a big claim that is unsupported, except for a single proximity ligation assay, with no controls. The caspase-9/Notch3 proximity ligation data shown in Figure 3g are very superficial and show a single cell, and no quantitation at the population level. How representative is this PLA assay? How many cells were counted? There are no controls for the PLA assay, did the authors try to do PLA with Notch3 and caspase-8 or any other caspase?

3. The IP based data in Fig 3h should be properly controlled by showing the accompanying IP blots to ensure that (a) caspase-9 was indeed immunoprecipitated and that (b) similar amounts of Notch3 and Notch3ICD were immunoprecipitated.

4. There is no such thing as a caspase-9-specific inhibitor in tissue culture, as claimed by the authors: "We first observed that caspase-9 is required for Notch-3-induced cell death as the caspase-9 inhibitor z-LEHD significantly inhibited cell death induced by Notch3 over-expression (Fig. 3c)". These peptides do not show specificity for individual caspases when used at micromolar concentrations (this has been demonstrated by Green and colleagues McStay et al., and by others). Here the authors used 5 uM which cross-inhibits all caspases. These data do not argue that caspase-9 is the most upstream caspase, the authors have not examined the effect of

caspase-9 knockdown or deletion on this pathway.

5. It is not clear to me why there is such effort devoted towards demonstrating that Notch3 is a caspase substrate (in Figure 3) if the authors do not demonstrate any significance for this cleavage event. Figure 3e has no loading controls, such as an actin re-probe.

Other comments

Suggested title change to: "Non-canonical NOTCH3 signalling limits tumour angiogenesis".

The manuscript needs quite heavy copyediting as the English expression is still quite poor in quite a few places.

Reviewer #3 (Remarks to the Author):

In general, the authors mostly satisfactorily addressed the concerns of reviewer 1. Specifically:

Specific comment 1: The reviewer asked for using additional cell lines to LLC1 to assess expression and tumor growth. The authors partially addressed this comment. For the expression data, they showed data from patient tumors. One additional cell line was used for the tumor growth studies, but it was a breast cancer cell line.

Comment 2: The reviewer asked for a longer period of tumor growth monitoring and presentation of the actual tumor volumes. This comment was satisfactorily addressed. The authors performed a longer experiments and showed the actual tumor volumes (Fig. 2a).

Comment 3: The reviewer asked for direct Notch3 staining to complement the LacZ staining. This issue was satisfactorily addressed in Figure 1.

Comment 4: The reviewer asked for greater magnification of the CD31 IHC sections and the use of immunofluorescence. This was partly satisfactorily addressed. The authors show greater magnification for one panel but did not use immunofluorescence.

Comment 5: The reviewer raised questions about studying whole vs transfected population in the apoptosis analyses and also suggested the use of 7AAD/AnnexinV staining. This comment was mostly satisfactorily addressed. The authors argue a high transfection efficiency and use AnnexinV/PI staining.

Comment 6: The reviewer asked for transfection experiments of S1-CterN3 in the presence or absence of DAPT and analysis of Notch target gene expression to prove that apoptosis is independent of canonical Notch signaling. The authors mostly satisfactorily addressed this comment. While they did not perform all the requested experiments they made a convincing case based on existing data and additional experimentation.

Comment 7: The reviewer asked for more mechanistic insight into how Notch induces apoptosis of tumor associated endothelial cells. The authors superficially addressed this comment by showing caspase dependence of the process. This is not real mechanistic insight.

Comment 8: The reviewer asked about the statistical significance of the reduction in CD31/Tunnel staining shown in Figure 2i. The authors satisfactorily answered this question in Figure 5e.

Comment 9: The reviewer asked for the quantification of apoptosis in the co-culture experiments.

The authors answered that no death was observed within the time frame of the experiment. Instead, they (satisfactorily) argued their conclusions based on data shown in Figures 3 and 4d.

Comment 10: The reviewer asked for a series of additional in vitro and in vivo experiments to prove that canonical Notch signaling is not involved in apoptosis in endothelial cells. The authors partly satisfactorily address this comment. They did not perform most of the requested experiments. Instead, they performed alternative experiments (Figure 5) and also important experiments showing the lack of the effects of DAPT on Notch target genes.

Referee #1:

“The reviewers consider the manuscript has been improved. We appreciate that the authors addressed most of the comments from the first revision. However there are still two major issues that we will like to see clarified if not experimentally at least in the text and discussion.”

We thank the referee for his/her kind comments.

“1. Related to the comment 5. The authors provided an experiment with caspase inhibitors to prove that the cell death observed is caspase dependent. How do the authors explain that the cell death is only partially inhibited either with Casp9 or pan-caspase inhibitors? “Is the NOTCH3 cleavage cause or consequence of the cell death observed in vitro? The lack of detection of the endogenous cleavage makes difficult to understand and relate the effects shown in HUVECs with the molecular mechanism proposed in the 293 cell overexpressing model.”

We do agree the initial data were suggesting only a partial inhibition with the caspase inhibitors. As also suggested by referee 2, we now have added an experiment with a specific silencing of caspase-9 which shows a stronger inhibition of cell death induced by Notch3 and we have more adequately demonstrated the importance of caspase-9 in Notch3-mediated cell death. However, at this stage we agree that we cannot exclude that the caspase inhibition by chemical inhibitors or gene silencing is associated with a switch to another kind of cell death. This is now mentioned in the text of the manuscript.

Regarding the point on caspase cleavage, this is an important question that remains completely opened for all dependence receptors and that unfortunately cannot be easily answered (as it asks to differentiate initial caspase cleavage from the amplification caspase cleavage). As this concern was also raised by referee 2 and as this caspase cleavage was initially just showed to strengthen the similarity with other dependence receptors but not to investigate the role of this cleavage per se in the death mechanisms, we have moved these data to Supplementary Fig.4ij

“2. The authors' statement in response to comment 9 is not correct - For the knowledge of the authors NOTCH3 floxed mice have been reported in literature i.e. <http://hyper.ahajournals.org/content/early/2016/06/13/HYPERTENSIONAHA.116.07694.abstract> <https://www.ncbi.nlm.nih.gov/pubmed/215366>”

We thank the referee for raising this important point. We had seen these manuscripts in the past and have looked at them again. However, and we apologize by anticipation if we have missed information in these papers, we cannot see the description of the notch3 floxed in these two manuscripts. In the first article "loss of Notch3 signaling in Vascular Smooth Muscle Cells Promotes Severe Heart Failure Upon Hypertension", although the title and the abstract might be a bit misleading, the authors compared Notch3 knock-out mice (developed by Thomas Gridley (Krebs et al., Genesis 37:139-143 (2003)) with conditional knock-out of RBPJK in smooth-muscle cells. They explain their choice page 2 of the article: "Because Notch3 conditional allele is not available, we therefore selectively deleted the RBPJK gene

specifically in the VSMCs in adult mice". In the second paper mentioned, the authors used the same Notch3 knock-out mice developed and characterized by Thomas Gridley (Krebs et al., Genesis 37:139-143 (2003). They use a conditional knock-out of Jagged-1 in VSMCs to demonstrate that Notch3 activation is dependent on this ligand. However if we have missed some information in Supplementary information (or have miss-read these papers), we would be happy to change our statement.

Referee #2:

"Here the authors suggest that Notch3 acts as a dependence receptor in endothelial cells and regulates tumour angiogenesis by regulating endothelial cell death via caspase-9 activation and apoptosis downstream."

We thank the referee for his/her comments and have tried to address his/her comments as followed.

"1. Given that the authors are claiming that Notch3 associates with caspase-9 and promotes activation of the latter, I am surprised that they did not attempt to assess the effect of caspase-9 knockdown or deletion (via CRISPR) on cell death induced by Notch3. Moreover, the claim that Notch3 associates with caspase-9 is based on very little evidence that is not properly controlled."

We fully agree with the referee and have tried to make the mechanistic data stronger. As suggested by the referee, we did a specific knock-down of caspase-9 in HUVEC which significantly inhibited the cell death induced by Notch3 (now shown in Figure 3d). We also improved the demonstration of the interaction of Notch3 with caspase-9. We show co-immunoprecipitation of Notch3 and caspase-9 with truncated version of Notch3 but not Notch1 and Notch2 (Figure 3e). We now also added immunoprecipitation of endogenous caspase-9 that interacts with Doxycycline-induced Notch3 (Figure 3f). Finally, we also did the proximity ligation assay with caspase-8 antibody, showing that the interaction of Notch3 is specific to caspase-9. Proper quantification of the interaction has also now been added (Figure 3g).

"2. How Notch3 promotes apoptosis is very obscure in the paper. The authors claim that Notch3 acts as a direct regulator of caspase-9 activation, but this is based on the most superficial of data. This is a big claim that is unsupported, except for a single proximity ligation assay, with no controls. The caspase-9/Notch3 proximity ligation data shown in Figure 3g are very superficial and show a single cell, and no quantitation at the population level. How representative is this PLA assay? How many cells were counted? There are no controls for the PLA assay, did the authors try to do PLA with Notch3 and caspase-8 or any other caspase?"

We apologize for the lack of strength of the initial mechanistic data that was a point the editor, while under review at NCB, mentioned as a point of relatively minor priority to investigate (i.e., "...do not think that providing detailed mechanism for how Notch3 triggers apoptosis in this setting would be strictly necessary"). However as suggested by the referee we have now strengthen the data on the role of caspase-9 in Notch3-induced cell death. We now have immunoprecipitation with endogenous caspase-9. Interestingly, we observed that N3ICD (which do not induce cell death) do not interact with caspase-9 suggesting the interaction between Notch3 and caspase-9 takes place only when Notch3 is anchored at the membrane (suggested also by the fact that the S1-Cter construct also interacts with caspase-9). We also performed the PLA assay using caspase-8 antibody and quantified the number of spots in 5 images of 100-120 cells. Data are now presented in Figure 3g.

"3. The IP based data in Fig 3h should be properly controlled by showing the accompanying IP blots to ensure that (a) caspase-9 was indeed immunoprecipitated and that (b) similar amounts of Notch3 and Notch3ICD were immunoprecipitated."

We now present an immunoprecipitation with endogenous caspase-9 and overexpressed Notch3 and N3ICD in Figure 3f, which clearly shows that N3ICD is not interacting with caspase-9. This new observation is consistent with the fact that N3ICD is not inducing caspase-9 activity (Figure 3h).

"4. There is no such thing as a caspase-9-specific inhibitor in tissue culture, as claimed by the authors: "We first observed that caspase-9 is required for Notch- 3-induced cell death as the caspase-9 inhibitor z-LEHD significantly inhibited cell death induced by Notch3 over-expression (Fig. 3c)". These peptides do not show specificity for individual caspases when used at micromolar concentrations (this has been demonstrated by Green and colleagues McStay et al., and by others). Here the authors used 5 uM which cross-inhibits all caspases. These data do not argue that caspase-9 is the most upstream caspase, the authors have not examined the effect of caspase-9 knockdown or deletion on this pathway."

We fully agree. We have modified the text accordingly and more importantly we now show that (i) the silencing of caspase-9 is inhibiting Notch3-induced cell death (Figure 3d) and that (ii) caspase-9 but not caspase-8 interacts with Notch3.

"5. It is not clear to me why there is such effort devoted towards demonstrating that Notch3 is a caspase substrate (in Figure 3) if the authors do not demonstrate any significance for this cleavage event. Figure 3e has no loading controls, such as an actin re-probe."

We initially did this to show the similarity of Notch3 with other dependence receptors. However as suggested this part has now been moved to Supplementary data.

"Suggested title change to: "Non-canonical NOTCH3 signalling limits tumour angiogenesis"."

The title has been changed as suggested.

“The manuscript needs quite heavy copyediting as the English expression is still quite poor in quite a few places.”

We apologize for the low quality of our English. The manuscript has been corrected by a native English-speaker and we hope that quality has improved. If not, the editing team will probably help us a lot.

Referee #3 (previously referee 1):

“In general, the authors mostly satisfactorily addressed the concerns of reviewer 1.”

We thank the referee for his/her very supporting comments.

We are grateful to the reviewers for their comments, which we believe have strengthened the manuscript. We believe that this manuscript provides new insights in the fields of oncology and apoptosis.

REVIEWERS' COMMENTS:

Reviewer #1 (Remarks to the Author):

The authors have addressed my last comments and I believe the manuscript is now ready for publication. Regarding the comment 2, after reviewing genetic mouse databases, I agree with the point that the Notch3-floxed mice are not available yet. This conditional KO is available in germ lines (e.g. Notch3^{tm1.1(KOMP)Vlcg}) that could be utilized for the generation of conditional KO mice. However, this would be beyond the scope of this manuscript.

Reviewer #2 (Remarks to the Author):

The authors have responses to the concerns satisfactorily